# Trends in Excess Winter Mortality (EWM) from 1900/01 to 2019/20—Evidence for a Complex System of Multiple Long-Term Trends

**DOI:** 10.3390/ijerph19063407

**Published:** 2022-03-14

**Authors:** Rodney P. Jones, Andriy Ponomarenko

**Affiliations:** 1Healthcare Analysis & Forecasting, Wantage OX12 0NE, UK; 2Department of Biophysics, Informatics and Medical Instrumentation, Odessa National Medical University, Valikhovsky Lane 2, 65082 Odessa, Ukraine; aponom@hotmail.com

**Keywords:** winter mortality, trends, season, estimated influenza mortality, pandemic influenza, aging, obesity

## Abstract

Trends in excess winter mortality (EWM) were investigated from the winter of 1900/01 to 2019/20. During the 1918–1919 Spanish flu epidemic a maximum EWM of 100% was observed in both Denmark and the USA, and 131% in Sweden. During the Spanish flu epidemic in the USA 70% of excess winter deaths were coded to influenza. EWM steadily declined from the Spanish flu peak to a minimum around the 1960s to 1980s. This decline was accompanied by a shift in deaths away from the winter and spring, and the EWM calculation shifted from a maximum around April to June in the early 1900s to around March since the late 1960s. EWM has a good correlation with the number of estimated influenza deaths, but in this context influenza pandemics after the Spanish flu only had an EWM equivalent to that for seasonal influenza. This was confirmed for a large sample of world countries for the three pandemics occurring after 1960. Using data from 1980 onward the effect of influenza vaccination on EWM were examined using a large international dataset. No effect of increasing influenza vaccination could be discerned; however, there are multiple competing forces influencing EWM which will obscure any underlying trend, e.g., increasing age at death, multimorbidity, dementia, polypharmacy, diabetes, and obesity—all of which either interfere with vaccine effectiveness or are risk factors for influenza death. After adjusting the trend in EWM in the USA influenza vaccination can be seen to be masking higher winter deaths among a high morbidity US population. Adjusting for the effect of increasing obesity counteracted some of the observed increase in EWM seen in the USA. Winter deaths are clearly the outcome of a complex system of competing long-term trends.

## 1. Definitions

Total winter deaths (TWD) = total deaths during the four winter months, December to March. This total is calculated as a rolling/moving total to detect winter deaths in the Southern hemisphere, near the equator and in those years when influenza outbreaks occur earlier or later than December/March, i.e., in the first calendar year of the time series start with the sum of January to April deaths, move forward by one month and recalculate, through to the end of the time series. TWD for each year is the 4-month period at which the rolling/moving total reaches a maximum.

Excess winter deaths (EWD) = total deaths during the four winter months minus half the deaths in the eight non-winter months. As a rolling/moving calculation as above. Hence for the first calendar year of the time series take the sum of September to December minus half the sum of January to August. EWD for each winter is the four months with the maximum value.

Excess winter mortality (EWM) = total deaths during the four winter months divided by half the total deaths during the eight non-winter months expressed as a percentage difference. As a rolling/moving calculation as above. EWM for each winter is the four months with the maximum percentage difference value.

Both TWD and EWD need to be adjusted for growth in deaths over time. EWM does not need such an adjustment. For example, between 7 January and 19 December deaths in the USA grew by 0.14% per month, which has a negligible impact on the EWM calculation.

## 2. Introduction

A previous study in *IJERPH* investigated the use of excess winter mortality (EWM) as a forensic tool [1]. The magnitude of EWM was shown to be greatly influenced by respiratory deaths and by a mix of other factors such as access to health care, home insulation (indoor temperature), and age at death [1]. All these factors are changing over time.

There has been considerable debate around exactly how many influenza deaths have occurred during various pandemics. The difficulty is that such deaths must be estimated and that different estimation methods give widely different answers [2,3,4,5,6,7,8]. Deaths due to influenza are a subset of EWM, and the high dependence of EWM on respiratory deaths [1] suggest that EWM may be able to shed additional insight into this issue.

Indeed, to the present there have been few studies on the long-term trends in EWM or investigation into other trends which may lie hidden in the EWM value.

This study seeks to conduct such analysis to see if EWM can be also used to detect the net effects of influenza vaccination or whether other trends could be confounding the association.

In this study EWM is calculated as the average deaths in the four ‘winter’ months (usually December to March) versus the average deaths in the previous eight ‘non-winter’ months (usually April to November) in the temperate countries [1]. To further generalize the applicability of this method the calculation is performed as a rolling/moving calculation, i.e., move forward one month and re-calculate. This allows EWM to be more accurately determined in winters with an early or late influenza outbreak, and for countries in the southern hemisphere or nearer to the equator [1]. This is a simple method which can be widely applied across multiple world countries but gives answers comparable to more complex methods, such as Serfling, etc. Deaths are total all-cause mortality, which avoids ambiguity in the assignment of cause of death, and influenza deaths are a subset of EWM—although a key contributor to EWM, i.e., estimated influenza deaths cannot exceed excess winter deaths [1].

## 3. Materials and Methods

### 3.1. Monthly Deaths

Monthly deaths are available from 1960 onward for European Union countries via Eurostat [9], and wider data for many world countries are available from 1980 onward via the United Nations [10]. Data for the USA from 1900 to 2004 were provided by Peter Doshi as part of his study on influenza mortality trends [11]. Missing monthly data for the five years 1905 to 1909 in the US time series were reconstructed from annual totals using the average value for the monthly deaths either side of these years as a baseline. Any excess of deaths in each year was added into the winter months such that the total deaths equaled the available annual total. Only one of these years shows high EWM. This was supplemented by monthly US data from 2005 onward [12]. Additional monthly data were obtained for US states [13]. Data for Denmark from 1901 onwards were obtained from Stats Denmark [14]. Data for Sweden were obtained from Stats Sweden [15]. Additional data from 1960 onward were obtained from Stats Singapore [16] and Stats Estonia [17]. Deaths in Puerto Rico are included in the total reported for the USA from 2018 onward. In this study they are excluded from the US total and the full time series for Puerto Rico is reported separately.

### 3.2. Adjusting EWM for Differences between Countries

The median EWM for world countries ranges from around 3% in Mongolia, 5% in Barbados, 6% in Russia and 7% in Singapore through to 25% in Portugal and Albania and 36% in Malta. The figure for the USA is around 11%. All countries are referenced to the USA by adjusting all EWM values such that the median EWM for each country is equal to that of the USA. For example, over a particular time frame the median EWM for country A may be 7% while that of the USA is 11%. All EWM values for country A are then multiplied by 11% ÷ 7% = 1.57. See Appendix A for adjustment factors.

In the USA, since the 1960s, EWM reached a minimum of 4.8% in the winter of 1973/74 and a maximum of 17.4% in the winter of 1962/63. Given that all EWM values in this study have been adjusted to USA-equivalent EWM, individual country values (after adjustment) below 5% have been adjusted to 5% while values above 20% have been adjusted to 20%. This prevents undue effects from outlying values which can arise in some of the smaller countries. In practice trimming makes little difference to the slope of the trends (below). The two studies on the effects of influenza vaccination used different date ranges and the median EWM adjustment factors for the same country can differ slightly. It is of interest to note that EWM in each country/region seems to occur as a dichotomous high/low state—hence the saw tooth behavior in Figure 1. The proportion of high/low EWM values seems characteristic of each country/region. The median value of EWM is therefore subject to further complex behavior (see Discussion).

### 3.3. Influenza Vaccine Doses Distributed per 1000 Population

Influenza vaccine doses distribution per 1000 population from the winter of 1980/81 to 2013/14 was from a series of publications by the MIV Study Group [18,19,20,21,22,23,24,25]. Additional doses distributed for the USA was from the CDC [26] divided by population [27]. A variety of northern hemisphere countries were chosen to reflect a range of low to high vaccination, i.e., former Soviet countries (lowest) to Japan/USA (highest). To accurately quantify the intercept at zero vaccination special emphasis was given to countries with very low vaccination levels. For the Netherlands and Finland, the doses of vaccine were extrapolated back to the winter of 1970/71, and the data for the USA were extended to 2019/20 given vaccine doses available from the US CDC [26,27]. The list of countries with ranges in doses distributed is given in Appendix A.

### 3.4. Influenza Vaccination Rates, Aged 65+

Influenza vaccination rates in those aged 65+ in the USA up to 2001/02 was from the study of Simonsen et al. [28], while more recent data were from the US CDC [29,30,31,32]. Data for 60 countries were from the OECD [33], European countries were from the study of Spruijt et al. [34], and other sources including the WHO Immunization Data Portal [35,36,37,38,39,40,41,42,43,44,45,46,47,48,49]. Since elderly vaccination rates are only widely available from 2000 onward a method based on a correlation between doses distributed (Section 3.3 above) and proportion elderly vaccinated in the overlapping period 2000 to 2013 was used to extrapolate backwards before 2000 based on doses distributed. In this method the nominal proportion elderly vaccinated calculated from the regression (above) was matched to the available trend from 2000 onward by adjusting the calculated value (from doses distributed) up or down to match the available trend in actual proportion elderly vaccinated. See Figure A1 in the Appendix B. From Figure A1 the maximum possible adjustment is 128% while the minimum is 78%. For example, a country distributes 200 does per 1000 population which corresponds to an average of 50% aged 65+ vaccinated. The actual value for age 65+ vaccinated can lie in the range 50% times 1.28 to times 0.78. Given that EWM does not appear to be highly sensitive to vaccination rates this adjustment was determined as follows. For example, at winter 2000/01 a country has 41% actual age 65+ vaccinated but has a value of 35% from the regression line in Figure A1. The adjustment factor for years prior to 2000 is then 41%/35% = 1.17 which is within the maximum and minimum limits. Some estimation is required since both doses distributed and actual age 65+ vaccinated tend to jump around, however, as stated this value does not require high precision, and before 2000 is generally low for many countries. This allowed an extended time series from 1988/89 onward. The list of countries with range in proportion elderly vaccinated is given in Appendix A.

### 3.5. Data Manipulation

Microsoft Excel was used to manipulate the data and produce the charts. Regression was performed using the Excel ‘add trendline’ function. 

## 4. Results

### 4.1. Long-Term Trends in EWM

To show a wider historic picture from 1900 to date, Figure 1 shows EWM for Denmark, Sweden, and the USA over the past 120 years. Over the winters 1959/60 to 2020/21 all three countries have nearly the same median EWM (Denmark 11.1%; Sweden and USA 11.4%) and hence no adjustment of the raw EWM is required. 

Due to their smaller size Denmark and Sweden have higher scatter due to Poisson randomness, i.e., based on average monthly deaths from 1960 to 2019, one standard deviation of random scatter in EWM is ±0.12% for the USA, ±0.58% for Sweden, and ±0.74% for Denmark—which makes an insignificant impact on the annual EWM. Several important observations can be made from Figure 1, namely:Both USA and Denmark show 100% EWM during the 1918/19 winter of the Spanish flu epidemic while Sweden had 131% EWM, i.e., EWM is directly influenced by, and sensitive to, influenza activity and virulence.While the EWM calculation for all three countries during the Spanish flu shows a maximum for the 4 months ending January 1919, the USA and Sweden shows maximum monthly deaths in October 1918, while maximum deaths in Denmark occurred in November 1918. In terms of the timing for higher Spanish flu deaths Denmark tends to be right skewed while the USA and Sweden are left skewed. This demonstrates the need for a rolling EWM calculation rather than a static one.In addition, EWM shows more complex trends, with a localized maximum around 1906–1907 (1900/01 to 1911/12) in Denmark and the USA, and a wider maximum centered around the time of the Spanish flu (1911/12 to 1930/31)—by coincidence Spanish flu occurred at a time of high baseline EWM. It should be noted that the emergence of Spanish flu precursor strains appears to have started somewhere around 1911 [50], and EWM appears to have correctly identified the timing of re-assortments culminating in the maximum at 1918 [50].After the Spanish flu, EWM then declined to reach a general minimum in the 1960s to 1980s which occurred in the very early days of the introduction of more widespread influenza vaccination.As an interesting observation there is no evidence that the four subsequent flu pandemics of the 1957–58 Asian flu, 1968–69 Hong Kong flu, 1977–79 Russian flu, or the 2009 Swine flu gave rise to unusually high EWM.There was another smaller peak for the three winters ending 1999/2000.

By way of comparison EWM is also an excellent tool to determine the exact magnitude of the COVID-19 pandemic and the winters of 2019/20 and 2020/21 in Figure 1 both include COVID-19 mortality. Despite widespread publicity the resulting EWM in these three countries is slightly higher but not remarkably so [51] (pp. 1–12). However, the national position in the USA conceals wide variation at state level with 76% EWM in New York at May 2020 and 52% EWM in South Dakota occurring at January 2021 [13]. Other countries suffered far higher mortality with North Macedonia seemingly highest in the world at 90% EWM ([51], p. 7). Given the differences in population age and social structure between COVID-19 and the Spanish flu plus additional international lockdowns, social distancing, wearing masks in public places and access to acute and critical care it is difficult to compare these two pandemics, however, on balance the Spanish flu seems to have had the highest mortality. Having established the long-term trends in EWM it is relevant to determine how the relative importance of winter may have changed over time.

### 4.2. Month in Which EWM Reaches a Maximum

Given the long-term trend to lower EWM in Figure 1 it is useful to see if the month in which the rolling EWM calculation reaches a maximum has changed over time. This is shown in Figure 2 with data from Denmark where the rolling EWM reached a maximum between April to June up to 1935, then switches to between March and April up to 1966, and beyond that predominantly occurs in March. Two exceptions occur in the winter of 1918/19 and 1957/58 where the maximum EWM occurs in January. The reasons for the seeming sudden transitions remains unknown.

Data for the USA show a similar transition from April in the early 1900s to March in the late 1960s. Sweden makes the transition to March in the late 1950s (data not shown). The transition for the USA is more muted than for Denmark because the USA lies further south than Denmark and has a far wider range in latitude. The situation in Sweden between 1851 and 1900 is more complex, with May to August up to 1860, and then March to May up to 1895 (data not shown). Far longer trends appear to be implicated with as unknown factors triggering the transitions.

As an additional observation, using the data on influenza deaths coded to ICD J10 and J11 from the study of Doshi [11] in the USA, reported “influenza” deaths appear to peak across 3 months in the early 1900s moving to 2 months in more recent years (data not shown). A transition appears to occur around the winter of 1967/68 (as also in Figure 2). Additional trends relating to influenza seasonality may lie concealed in the time trends.

### 4.3. Winter Deaths Are Higher before the 1970s

Figure 1 also suggests that winter deaths were higher relative to summer deaths in the early 1900s. This is explored for Denmark in Figure 3 where the deaths in each month have been adjusted to equal days per month, and then to the 1995 equivalent number of maximum annual deaths. 

The *Y*-axis is truncated at 8400. Deaths per month merge over time with June and July showing the least change. The seven months June to December show an increase over time, while the five months January to May reduce over time. February, November, and December are the most volatile. Not only have monthly deaths converged but relative volatility may have shifted over time. Hence relative to the period around the time of the Spanish flu deaths have been shifted out of the (moveable) “winter” (January to May) into the “non-winter months”. These trends seem to stabilize from the 1970s onward. This explains the reduction in EWM over time seen in Figure 1. Once again, complex trends lie behind the calculated EWM which are poorly understood.

However, it is pleasing to note that the definition of “winter” in the EWM calculation as covering 4 months is sufficiently wide to cover all the obscure trends since the 1900s and earlier.

### 4.4. Similar Trends Are Seen in World Countries since the 1960s

It is useful to see if these trends are replicated wider than just Denmark, Sweden and the USA and Figure 4 shows data for a wide selection of countries with local EWM converted to the USA-equivalent as per the median-EWM method in Section 3.5. 

In Figure 4 the maximum and minimum rolling EWM is shown for the 25 countries with continuous data from 1960 onward, while the upper and lower quartile is shown for a far wider group of up to 113 countries and 25 states/provinces for the years in which data are available. EWM data for countries in the southern hemisphere were shifted forward by six months to give the northern hemisphere equivalent.

As can be seen there is no evidence for unusually high EWM during the three influenza pandemics occurring over this time frame. The early years of this chart pick up on the end of the downward trend seen in Figure 1. Considerable spatiotemporal variation can be seen in all years, although in general countries move up and down together. Hence the data in Figure 1 is more widely applicable than just Denmark, Sweden, and the USA. 

Figure A2 (Appendix B) shows the proportion of excess winter deaths (EWD) due to “influenza” (ICD codes J10 to J11) in the USA from the winter of 1900/01 to 2003/04. Data are from the study of Doshi [11]. As can be seen this proportion reaches a maximum of 70% during the Spanish flu pandemic, but regularly goes as high as 50% up to the mid-1940s. More widespread availability of penicillin and other antimicrobials from 1945 onward could be one possible reason for the shift down after the mid-1940s. There are complex long-term undulations but no indication that the epidemic years after the Spanish flu have higher deaths than seasonal influenza. This will be covered further in the Section 5.4.

### 4.5. EWM Correlates Well with Calculated Influenza Deaths

Given that influenza was the major winter pathogen over these years it is useful to see if EWM correlates with estimates of influenza deaths. Such a relationship is demonstrated in Figure 5 using data from Denmark where the FluMOMO European-wide methodology have been employed for the estimated influenza deaths. This is the most advanced methodology which uses weekly deaths after adjustment for the effect of cold temperature extremes, as per the FluMOMO methodology which is available as open-source software [6]. Three different methods are used with influenza-like-illness (ILI), etc., described in the study of Nielson et al. [52].

The line for the Goldstein Index probably gives the most reliable estimate of influenza deaths. However, all three lines give good correlation and EWM explains > 85% (as R-squared) of the observed variation in estimated influenza deaths. As similar chart is available for data from Canada using data from the study of Schanzer et al. [53] and is given in the Appendix B as Figure A3.

It can therefore be concluded that in all influenza pandemics, other than the Spanish flu, influenza deaths are about the same as expected from seasonal influenza. See discussion regarding role of antigenic distance in Section 5.4.

### 4.6. Is the Effect of Influenza Vaccination Detectable Using EWM?

Given that influenza was the major winter pathogen over the period of this study it is interesting if rising international influenza vaccination levels are associated with any change in EWM over time. This issue was addressed in two ways. In the first EWM was plotted against influenza vaccine doses distributed per 1000 population (see Figure A4 in the Appendix B). Data were available from 1980 to 2013 and only northern hemisphere countries were included. A slight trend to higher EWM was seen, however, there was no statistical difference from the null hypothesis, namely, no change (*t*-value > 10). A list of countries with associated data ranges is given in Appendix A.

In the second method EWM was plotted against proportion age 65+ vaccinated and this is shown in Figure 6. Data were available from 1988 to 2019 for 97 northern and southern hemisphere countries. A slight negative slope was seen, however, there was no statistical difference to the null hypothesis (*t*-value > 10). A list of countries with associated data ranges are given in Appendix A.

During the period of the study there have been trends in multimorbidity, polypharmacy and obesity which are acting to oppose influenza vaccination and these and other trends are discussed in Section 5.6.

### 4.7. Is Influenza Vaccination Masking a Trend to Poor Health in the USA?

Figure 6 could be concealing a trend upward in EWM in some countries. Given the fact that the USA has among the highest levels of obesity, diabetes, and cancer in the developed world (see Section 5.6) we proposed that EWM may be trending upward over time in the US and that influenza vaccination may be masking the extent of this trend. This is illustrated in Figure 7 where the maximum possible effect of influenza vaccination has been estimated to be a 10% (percentage point) reduction in EWM for a 100% vaccinated population at 100% VE. However, in the USA the average vaccine effectiveness (VE) for persons aged 65+ has only been 40% [54], and so the blue dots represent the likely EWM in the absence of any vaccination. To adjust for the effect of no vaccination elderly vaccination rates and VE were matched with date and the estimated effect applied.

As can be seen there is indeed a trend upward in EWM since the 1960s (0.02% per annum) and that this trend would be around 3-times higher (0.07% per annum) if there were no influenza vaccination. The two dotted lines are 5th order polynomial curve fits (see Section 5.7) to show that the fundamental trends are probably far more complex, however, these lie around the approximate upward linear trends indicative of declining population health over time. 

The US population is seemingly paying a high price for lifestyle choices fueling obesity and related morbidities, which in terms of excess winter deaths is being masked by widespread influenza vaccination. Discussed in Section 5.6.

### 4.8. Obesity and Median EWM among American States

To investigate if obesity was linked to higher EWM, obesity levels for persons aged 65+ in American states in 2000 were compared to the median EWM (2007/08 to 2019/20) for each state [13]. Median EWM was first adjusted for latitude using a second order polynomial [1]. Data for Singapore were included as a wealthy country close to the equator. Using the latitude adjusted EWM, an assumed linear relationship between EWM and obesity gave a slope of 0.067 (all states) or 0.064 after removing outlying states with high/low EWM. See Appendix A.

The wider context to the slope is that between 2000 and 2018 obesity among those aged 65+ rose at 0.63% per annum [55]. 

The underlying effect of increasing obesity allows Figure 7 to be re-expressed in terms of the trend in EWM as a function of proportion elderly vaccinated as shown in Figure 6. This is shown in Figure A5 in the Appendix B. After adjusting for the effect of obesity an apparent increasing trend in EWM with vaccination is transformed into a near-stationary trend in EWM as vaccination increases. Since obesity is only one of several factors increasing EWM the ultimate trend will have a lower slope than 0.0028 (after obesity adjustment). See discussion in Section 5.6.

## 5. Discussion

### 5.1. Long-Term Trends in EWM

To our knowledge this is the first study looking at the long-term trends in EWM. Figure 1 shows evidence for long-term cycles with a minimum in the early 1900s, a maximum around 1906 and then another minimum around 1911. The work of Smith et al. [50] has identified that Spanish flu precursor strains appear to have emerged somewhere around 1911, and EWM appears to have correctly identified the timing of re-assortments culminating in the maximum at 1918 [50].

Note that after the Spanish flu, EWM declines during the pre–antibiotic era (Figure 1). This decline is most likely to be due to improvements in sanitation, refuse collection, health literacy, improved nutrition, and other general improvements in public health, building standards [56,57], and finally the eradication of widespread respiratory tuberculosis via BCG vaccination after World War 2 [58]. The three countries then reach a broad minimum in EWM between 1961 and 1982. 

The curious trends in the timing and magnitude of winter deaths observed in Figure 2 and Figure 3 are reflected in the timing of maximum EWM revealed by the rolling EWM calculation. We are unable to offer any explanation other than to point out that such trends are occurring but seem to have reached an asymptote from 1967 onward. This at least removes one unknown confounding factor from the analysis conducted in Figure 6 and Figure A3 which commence from 1980 onward. 

Over the period 1860 to 1995 in Sweden a reduction in the difference between summer and winter deaths was also observed in older people in the study of Ledberg [59]. The main part of the reduction in seasonality was observed to occur between 1870 to 1970, the latter date being somewhat close to the 1980 date in Denmark when the shift reached an asymptote. 

About 40% of increased life expectancy at age 60 could be attributed to the decrease in seasonality. In 1860 most of the population lived in the country engaged in agriculture and forestry while by 1970 Sweden was a prosperous country with a highly developed welfare system and most of the population living in cities [59].

### 5.2. EWM Is Sensitive to Influenza Deaths

Figure 5 and Figure A2 suggest that EWM is explaining >85% of the observed variation in estimated influenza deaths, and hence, there is every reason to believe that EWM has correctly identified the fact that influenza pandemics since the Spanish flu have not generated any excess deaths over and above that from seasonal influenza as seen in Figure 1 and Figure 4.

Additionally note that somewhat paradoxically EWM is highest midway between the poles and the equator [1]. This is simply because countries near the equator experience minimal temperature variation and those nearest the poles are better prepared for winter, i.e., the issues are wider than just external winter temperature. This has wider implications to the estimation of global influenza deaths both during pandemics and for seasonal influenza which are discussed in the next section.

### 5.3. The EWM Adjustment Factor

As a rule, EWM reaches a maximum at approximately midway between the equator and the poles (45°)—for example, Spain, France, Croatia, Serbia, Romania, Ukraine, northern Italy, and the most northern US states (northern hemisphere), and New Zealand, and parts of Chile and Argentina (southern hemisphere) [1]. The USA has a generally low international value of EWM. One of the most prominent reasons is its sheer size and range in latitude (24 to 49° N) which means that the spatiotemporal variability of EWM at the state level is moderated. The resulting variation in timing and magnitude leads to offsets creating the appearance of low EWM at national level. EWM adjustment is therefore required to bring other countries to a comparable value to that for the USA—as the country with the longest time series of both EWM and influenza vaccination. 

The adjustment factors for the countries are a complex mix of geographic size (as for the USA), latitude, altitude, microclimate, housing standards, access to health care, population age structure, and relative wealth and amount of fuel poverty for winter heating [1]. All of these combine to determine the median EWM for each country. For example, in the USA the three adjacent northern Rocky Mountain states all have unusually low median EWM with respect to their latitude, suggesting a potential role for altitude and resulting adaptation of buildings to withstand cold. The most obvious adjustment, namely using median EWM, has been used in this study. 

It is acknowledged that the use of median EWM has limitations and Appendix A [60] shows long-term trends in EWM for four countries. The difference in the trends relative to the USA are unlikely to have a major impact on the scatter seen in Figure 6 and Figure A4.

### 5.4. Evidence for Seasonal Levels of Influenza Deaths during Subsequent Pandemics

This anomaly between estimated pandemic deaths and recorded influenza deaths was first noted by Doshi [11]. Put simply, influenza deaths are a subset of EWM and reported ‘higher’ pandemic deaths may be an artefact of the methods used to estimate influenza deaths [2,3,4,5,6,7,8,61], or due to reporting bias where deaths during the pandemics are not compared to non–pandemic years.

It has been proposed that the very high mortality seen during the Spanish flu pandemic arose from previous exposure during the 1889–90 Asiatic flu pandemic. This led to a very high mortality peak at the exact age of 28 [62]. Original antigenic sin or antigenic imprinting during the earlier Asiatic flu then led to immune dysregulation during exposure to a highly dissimilar variant/clade during the Spanish flu [62]. 

This can work in the other direction, and two notable examples exist where pandemic influenza did not increase deaths due to antigenic imprinting. The 1977–79 Russian flu pandemic was antigenically similar to strains circulating in 1947 to 1956 [63,64], and antigenic imprinting created pre–existing immunity and low deaths; as demonstrated in Figure 1 and Figure 4. The same holds for the 2009 Swine flu pandemic [65,66] in which there were very few elderly deaths due to childhood exposure to similar strains, as also seen in Figure 1 and Figure 4.

The issue is not the ‘pandemic’ designation but antigenic distance from previous lifetime exposure [67,68,69,70,71,72]. As demonstrated in Figure 1, seemingly only the Spanish flu met the antigenic distance criteria. The problem seems to be one of selective reporting of pandemic deaths without comparison to other years, of which 2017/18 is a non-pandemic example where influenzas type B were unusually high, there was a vaccine mismatch due to a shift in dominant clades [65], and consequently VE was generally low [73].

Regarding the issue of pandemic influenza deaths from Figure 1 it should be noted that other studies have commented upon the apparent absence of higher influenza deaths during pandemics compared to seasonal influenza [74]—Spanish flu excepted. It was proposed that the wider publicity surrounding pandemics may have led doctors to code more hospital admissions and deaths to influenza/pneumonia thereby artificially inflating the estimates of influenza deaths [74].

Some comment needs to be made regarding the study of Peter Doshi [11] which investigated trends in recorded influenza mortality (ICD codes J10 and J11). Doshi was fully aware that ICD codes J10/J11 underestimate total influenza mortality, however, his study used these codes as a common denominator to explore the trend over time, i.e., there is no a priori reason for doctors to record fewer J10/J11 as cause of death over time. Trends in recorded influenza deaths in his study and observed for EWM in Figure 1 of this study both decline over time, and do not show undue deaths during pandemics after the Spanish flu. This point was further emphasized in Figure A2.

Also note that EWM is highest midway between the poles and the equator and reaches an absolute minimum at the equator [1]. Countries nearest the poles are better prepared for winter [1], i.e., the issues are wider than just external winter temperature. (see Section 5.6).

International estimates of influenza deaths do not account for the role of latitude in moderating EWM and influenza deaths, and overestimation of deaths is likely. Most of Africa is equatorial or sub-equatorial and will have a low EWM (although monthly data are absent for most of Africa and deaths are estimated). As demonstrated in Figure 5 the method developed in FluMOMO using the Goldstein Index probably gives the most reliable basis. We note that the method currently used in the USA by the CDC appears to overestimate influenza deaths by around 14% to 30% (See Appendix A [75,76,77,78,79,80] in the Appendix A. The USA is so large and covers such a wide range in latitude that monthly or weekly deaths in each state can be converted to EWM or a FluMOMO equivalent.

### 5.5. What Constitutes a “Good” Estimate of Influenza Deaths?

As a general rule, any method subject to over-estimation will have a low magnitude point of intersection through the *X*-axis in a plot of estimated influenza deaths versus EWM—as seen for the three methods in Figure 5. An appropriate point of intersection through the *X*-axis was the rationale for the “suggested flu deaths” line shown on Appendix A which is a best-case scenario giving around 14% over-estimation. The point off intersection on the *X*-axis is around 7%, which is low compared to Figure 5. A point of intersection around 8% gives 30% over-estimation—see below. Recall that these figures are specific to the USA.

The point of intersection will lie somewhere higher than the minimum EWM determined for each country. The point of minimum EWM will occur when:Influenza infection is low to near absent.Other winter pathogens are also less activeThe winter is exceptionally mild, i.e., warmOther factors are suppressing airborne pathogen spread

Hence, close to zero influenza deaths will occur at a range of lowest EWM values. For example, in the 120 years since 1900 the range for the six lowest EWM for the three countries in Figure 1 are Denmark (3.1% to 6.8%), Sweden (4.9% to 6.9%), and USA (4.8% to 7.4%)—mostly clustered in the 1960s and 1970s when EWM in Figure 1 went through a minimum. The point of intersection should therefore lie somewhere above 7% for these countries as in Figure 5 where the most likely value for Denmark is around 8.5%.

### 5.6. EWM and Influenza Vaccination—Competing Underlying Trends

Most advanced economies start to increase influenza vaccination during the 1980s [18,19,20,21,22,23,24,25]. According to the WHO by 2000 only 20 countries had policies for seasonal influenza vaccination, rising to 91 in 2010 and 124 in 2020 [81], hence, higher levels of vaccination in the elderly only generally occur beyond 2000. In 2000 elderly vaccination rates ranged up to 76% in the Netherlands (the majority below 65%), to a maximum of 86% in 2019 for South Korea (the majority below 70%) [31,33].

Influenza vaccination is known to reduce both hospitalization and death in the elderly [82] and should act to reduce EWM. Several studies have estimated that influenza vaccination should make a detectable effect upon EWM [28,83]. The fact that Figure 6 and Figure A4 do not demonstrate any downward trend raises the question as to whether the effect of influenza vaccination is being counterbalanced by other trends. Figure 7 investigated this possibility using EWM data from the USA, which is possibly the developed country with the worst morbidity trends. It was proposed that rising morbidity was causing EWM to rise by an upper limit of 0.07% per annum (in the absence of influenza vaccination) and that increasing influenza vaccination over time led to the observed lower value of 0.02% per annum increase. 

The issues around rising morbidity is explored in Table 1 where increases in multimorbidity, polypharmacy, obesity, Alzheimer’s and dementia, and age at death will all contribute to pressure to increase EWM since the 1980s. Several factors contributing to lower EWM are also discussed.

Age at death is probably the greatest contributor to higher EWM and in England and Wales the proportion of persons dying at age 65+ has risen since 1980 from 82% to 88% (in 2019) in females and from 71% to 82% in males [125]. Age will also be serving as a proxy for many of the morbidity issues. Since 1980, average age at death over the age of 65 has risen from 76 to 81 in males and 80 to 85 in females [125]. Age will be intertwined with multimorbidity, polypharmacy, etc.

These forces are opposed by improvements in home insulation, reduced smoking prevalence, improvements in influenza vaccination technology and more widespread influenza vaccination in the elderly.

There have not been any studies looking at how all the factors may (synergistically) combine to change EWM over time, and thereby diminish the ability of influenza vaccination to reduce EWM. All these trends will be country specific, and it is suggested that the analysis in Figure 6 and Figure A4 be repeated for each country to disentangle the multiple trends.

The role of temperature is more problematic. The temperature for minimum deaths in each location is very close to the most frequent temperature experienced in that location [126]. Deaths respond far more to extreme temperature events [127]. Hence, it has been noted that although deaths increase as it gets colder, temperature alone only explains a small amount of the variation in EWM between years [128]. There are complex interactions between air pollution (PM10), temperature and influenza on all–cause, respiratory, and cardiovascular mortality [129]. Each of these variables operates both alone (PM10 mainly affects cardiovascular, and influenza affects mainly respiratory) and in combination with additional specific interactions between influenza–PM10 for cardiovascular mortality and between influenza–temperature upon all–cause mortality [129]. The relationships are complex and will mostly affect those living in large cities—which usually make a major contribution to total deaths and hence the whole country value of EWM.

Once again, the complex relationships between temperature/air pollution/influenza will be country specific.

Indoor temperature is a known major factor in EWM [88,89,90,91,92,93,94], which suggests that it is not influenza per se, but influenza infection exaggerated by poorly maintained indoor temperature which is a contributory factor in deaths. It is extremely difficult to disentangle the two.

### 5.7. Undulations in the Long-Term Trends

Figure 1 noted a local maximum in EWM around the winter of 1999/00. To explore greater complexity in the long-term trends Figure 7, Figure A5 and Appendix A included polynomial curve fits. In the absence of any mechanistic understanding behind the undulations the polynomial curve fits are placed on the Figures to be illustrative. The choice between 4th and 5th order polynomial is arbitrary and is to avoid seeming large divergence in the more recent data.

The undulating trend for the USA in Figure A5 may have resolved the question as to why the study of Simonsen et al. [28] failed to detect a reduction in winter mortality in the USA during a period of rapidly increasing elderly influenza vaccination after 1980 and up to 2000/01 (when their study finished). By chance, this period corresponds to the upward part of the cycle in EWM observed in Figure A5. Note that in Figure A5 elderly vaccination was around 25% for winter of 1986/87 (minimum point in the cycle of EWM) and had risen to 65% by 1999/00 (maximum point in the cycle of EWM). Hidden complexity in the long-term trends acted to confound the results of that study. Adjustment for the cyclical nature of EWM would presumably have yielded a result that rising influenza vaccination was indeed beneficial.

Analysis of the trend for the USA in Figure 1 suggests that there may be another local maximum around 1937 (data not shown). Analysis of data from Sweden between 1851 and 1900 also shows evidence for further undulations (see Appendix A). The trends are far more complex than first appears and may well involve influenza clades and sub-clades plus non-surface coat genetic mutations in addition to other factors.

The saw-tooth behavior in Figure 1 also suggests that the data show high/low behavior. An example of this is given in Appendix A for American states for the 13 winters up to 2019/20. Low EWM winters are more tightly clustered in terms of the range in EWM, and the relative frequency of High/Low appears to be a characteristic of each state. As it were, more evidence for complex behavior.

### 5.8. Strengths and Limitations

The major strengths of this study are that very long-term trends have been investigated for the first time using a method which can be applied to international data. Apart from the Spanish flu pandemic the absence of higher pandemic influenza deaths has been established from multiple perspectives. The ability of influenza vaccination to reduce EWM was investigated with two large datasets, however, these lacked information on the relative effects of the opposing forces. The role of competing trends was highlighted but quantifying all these trends was beyond the scope of the study. However, the trend in obesity in the USA was alone sufficient to explain a significant part of the overall upward trend in EWM observed in the USA.

The method based on median EWM to adjust all countries to the US equivalent could potentially suffer from country-specific trends in EWM which are different from those in the US.

It is possible that additional complexity between countries lies at annual level.

## 6. Conclusions

EWM is a simple but powerful way of investigating the role of multiple and complex trends over time. ‘Winter’ has seemingly moved from the four months ending April/May in the early 1900s to ending March in more recent times. Deaths have seemingly been shifted out of winter as time has progressed, but in the three developed countries studied this had seemingly reached an asymptote by the mid-1960s. Higher influenza deaths during pandemics (except the Spanish flu) have been discounted on multiple fronts, and the role for antigenic distance was highlighted. 

Influenza deaths form part of the economic justification for influenza vaccination. It is recommended that researchers and public health agencies use and report both EWM and FluMOMO to estimate influenza deaths and to cross validate each against the other. Any numbers produced should be accompanied by a clear explanation of likely bias. 

For example, in the study of Rosano et al. [130] regarding influenza mortality in Italy the estimated value in 2015/16 of 15,801 influenza deaths is a clear outlier given that EWM was only 9.5% in that year. Such a low figure would only imply around 1621 influenza deaths. This discrepancy would lead to further investigation. Note the intersection point on the *X*-axis in the Rossario et al. study [130] excluding 2015/16 is around 8% EWM. This is plausible given that the EWM in 2015/16 was the second lowest in Italy since 1990/91.

The seeming lack of effect against EWM due to rising influenza vaccination has raised the possibility that increasing aspects of elderly morbidity are acting to obscure any trend. The exact contribution of the opposing trends needs to be quantified as does the issue as to whether these trends are additive or synergistic. The mechanistic basis for the long-term undulations in EWM require investigation.

The factors contributing to EWM are showing high complexity and complexity theory dictates that further unexpected outcomes lie concealed in the data [131].

## Figures and Tables

**Figure 1 ijerph-19-03407-f001:**
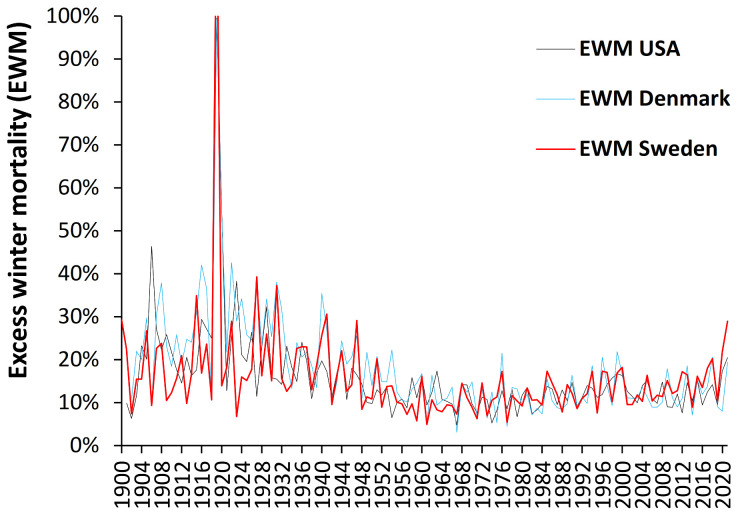
Excess winter mortality (EWM) over the past 120 years in Denmark, Sweden, and the USA. Note the *Y*-axis has been truncated at 100%.

**Figure 2 ijerph-19-03407-f002:**
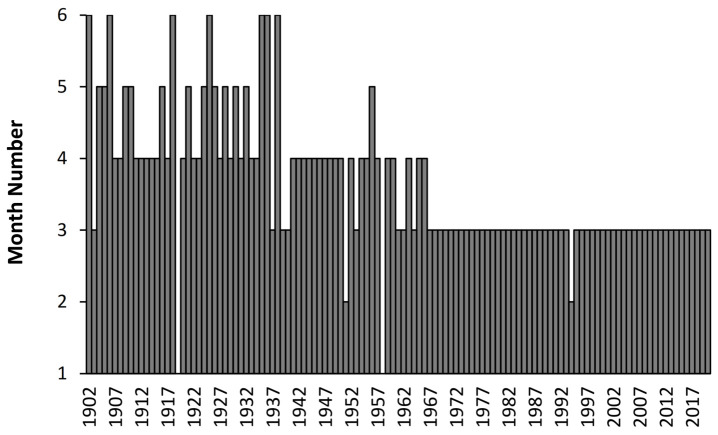
Month of the year at which the rolling EWM calculation reaches its maximum value in Denmark, month 1 = January, etc.

**Figure 3 ijerph-19-03407-f003:**
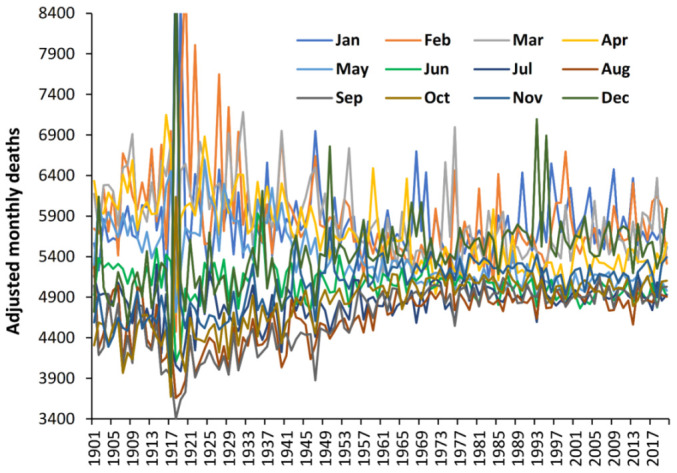
Trend in adjusted monthly deaths in Denmark. Monthly deaths first adjusted to give equal days per month and then adjusted to give an annual total equal to the maximum achieved in 1995.

**Figure 4 ijerph-19-03407-f004:**
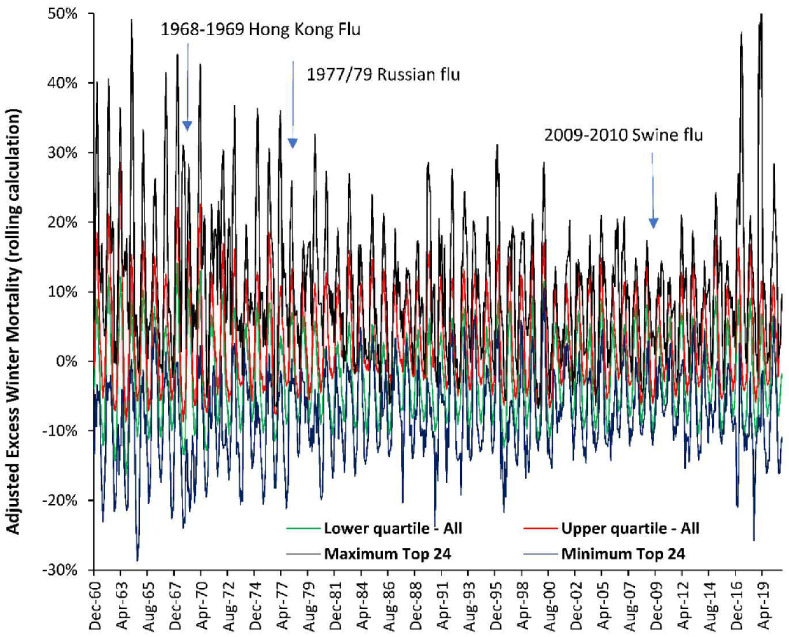
Trend in adjusted EWM for 25 countries (1960 to 2020) plus up to 113 countries and a further 34 states/provinces from Australia, Canada, and Germany (1980 to 2020). Countries are from both the northern and southern hemisphere.

**Figure 5 ijerph-19-03407-f005:**
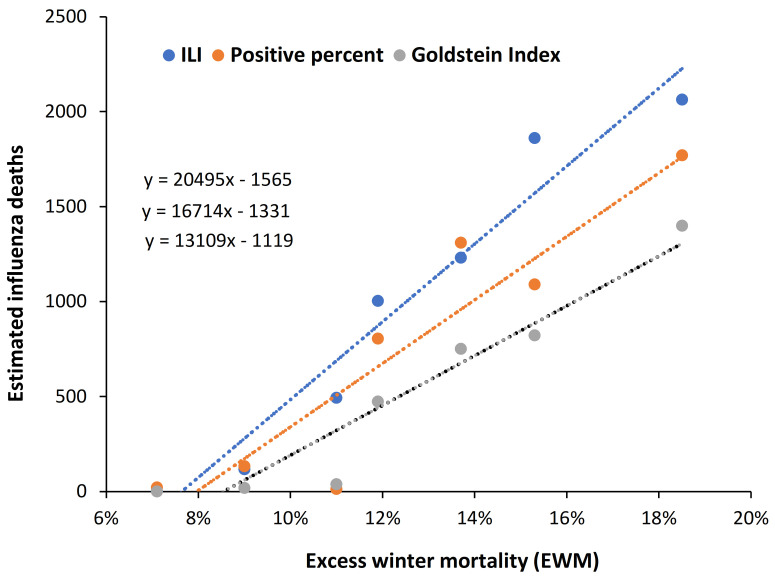
Three different methods for estimating influenza deaths in Denmark, 2010/11 to 2016/17. Influenza deaths are from the study of Nielson et al. [52]. FluMOMO data were adjusted for excess deaths arising from periods of very cold weather. The Goldstein Index method is considered the most reliable method. R-squared for the 3-methods ranges from 0.851 (Positive percent), 0.9178 (Goldstein) to 0.9482 (Influenza-like-illness).

**Figure 6 ijerph-19-03407-f006:**
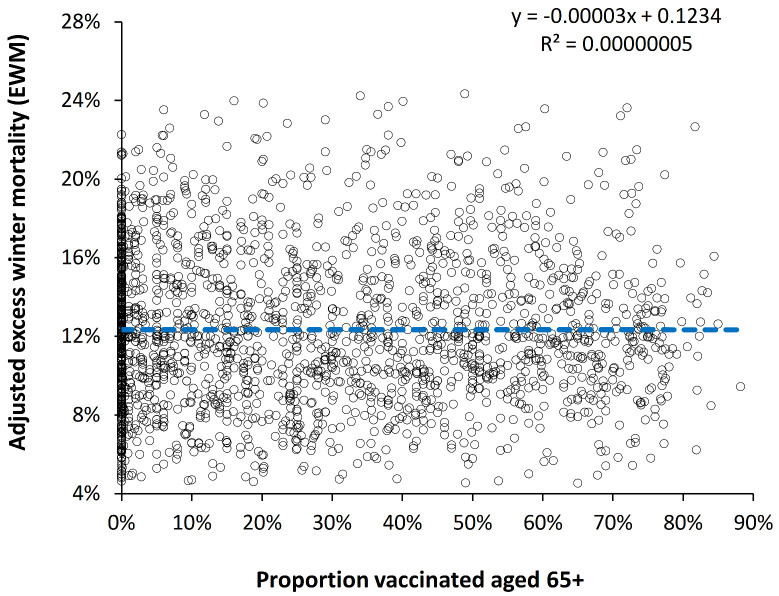
Adjusted EWM versus proportion aged 65+ vaccinated in 97 countries, 1988/89 to 2019/20.

**Figure 7 ijerph-19-03407-f007:**
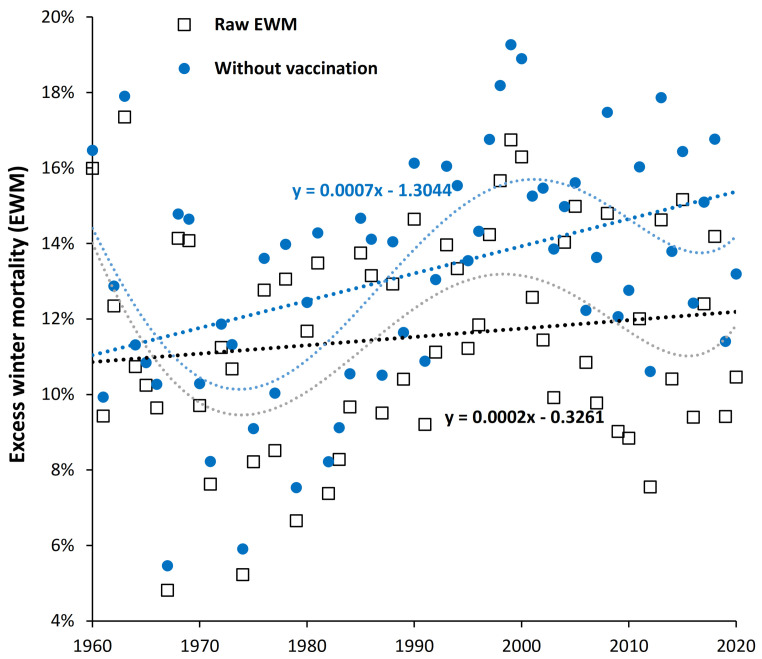
Trend in excess winter mortality (EWM) in the USA from 1960 to 2020, with and without correction for the effect of influenza vaccination. Vaccine effectiveness before 2003/04 estimated at 40%, actual VE values for 2003/04 onward [54]. Percent elderly vaccinated increases from around 15% in the early 1960s [28] rising to 67% to 70% from 2001 onward [29,30,31,32,33].

**Table 1 ijerph-19-03407-t001:** Factors increasing or decreasing EWM with time.

Factors Increasing EWM with Time	Factors Reducing EWM with Time
Multimorbidity—Levels of basic and complex multimorbidity have been increasing over time [84,85]. Multimorbidity is associated with diminished response to influenza vaccination [86] and risk of influenza mortality [87].	Home insulation—this is a major contributor to reductions in winter hospitalization and mortality in some countries [88,89,90,91,92,93,94]. The greatest benefit will occur in countries mid-way between the poles and equator which tend to have housing suited to summer rather than winter [1].
Polypharmacy—Polypharmacy in the Netherlands and USA had more than doubled in the interval 1999 to 2014 [95]. Certain pharmaceuticals and polypharmacy alter the immune response to influenza vaccination [96]. Polypharmacy has been found to be a risk factor for hospitalization, and death from COVID-19 [97] and for pneumonia admissions for nursing home residents [98,99].	Increased access to health care (critical care, antibiotics, antivirals, etc.) and wider public health measures [1]—this will mainly apply to the less developed countries.
Obesity—has been increasing over time [100]. It creates systemic inflammation, reduces B-cell function [101,102], generates auto-immune antibodies during infection [103,104], and interferes with influenza vaccination efficiency [103,104]. As demonstrated in Appendix A and Figure A5 the effect of the trends in elderly obesity alone has the potential to explain a moderate proportion of the gap between the lines in Figure 7.	Reduced smoking prevalence—smoking leads to inflammation and is a risk factor in influenza mortality [105,106,107].
Alzheimers and dementia—Incidence increases exponentially with age [108] and are a significant risk factor for influenza mortality [109].	Improvements in influenza vaccine technology such as cell versus egg grown vaccines, adjuvants, etc. [110,111].
Diabetes—Incidence increases with age [112] and is a significant risk factor in influenza mortality [113]. In the USA persons aged 65+ experienced the greatest increase in the incidence of diabetes since 1988 [112]. However, mortality is reduced by influenza vaccination [114,115].	Influenza vaccination in the elderly—increased vaccination will lead to lower influenza deaths [82], especially in years with a high VE.
Cancer—Cancer incidence increases with age [116] with incidence especially high in the US [116]. Cancer survivors are at far higher risk of influenza mortality [117]. Mortality is reduced by influenza vaccination [118]	
Air pollution (especially in large cities with population growth)—Air pollution is well recognized for its ability to increase systemic inflammation [119], alter aspects of immune function [120], increase incidence of ILI [121], and increase the proportion of persons infected with influenza [122]. It also interferes with influenza vaccination efficiency and is a risk factor for influenza mortality [123]. In children, air pollution has an adverse effect on lung function which is moderated by influenza vaccination [124].	
Longeivity or increasing age at death—EWM increases with age at death, however, difficult to assess over many decades as chronological age is not a good measure of biological or epigenetic age [1].	

## Data Availability

All data is publicly available. Data compiled for this study is available on request from Rodney Jones, email: hcaf_rod@yahoo.co.uk.

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
