# Peer review of "Trends in Excess Winter Mortality (EWM) from 1900/01 to 2019/20—Evidence for a Complex System of Multiple Long-Term Trends"

_ijerph, 2022, doi:10.3390/ijerph19063407_

Round 1

Reviewer 1 Report

The present work analyses the time evolution of excess mortality in different countries for more than a century. Such a study is of great interest now because it provides material for further exploration of the effects of the recent COVID19 pandemics as providing a reference given previous pandemic evens such as Spanish flu, Asian and Russian flues, etc. 

The analysis is demonstrative and well-discussed; thus I recommend accepting this work after some revision aimed at the correction of minor issues:

  1. Line 30: why Southern hemisphere? The countries considered are in the Northern hemisphere, probably it is a misprint. 
  2. Fig. 4: the position of an arrow marking Russian flu is misplaced, it denotes 1997/98, while the correct position is 1977/79 (as correctly noted in subsection 4.4 and throughout the text).
  3. Fig 1A: the data looks like a curve with saturation (this also follows from the discussion in the text), therefore fitting by a high-order polynomial is better to replace with some saturating curve like exponential, sigmoidal, or something like this.

Author Response

Many thanks for your time and comments. You will note that additional data has been included in Figure 1 for Sweden, which gives the same result as for Denmark and USA but a higher EWM during Spanish flu. Data from the USA also enabled an estimate of the effects of obesity on EWM.

Otherwise to address your points.

  1. This has been clarified. Different figures use different lists of countries. Figs 4 and 6 use countries from both hemispheres while Fig A4 was restricted to northern hemisphere countries. Figure captions have been ammended to make this clear.
  2. Thank you for spotting this error! The arrow and dates have been changed.
  3. A section dealing with the undulatory nature of the trends has been added and some analysis of data from Sweden going back to 1850. Whether this is an asymptote remains to be seen - maybe we will know in 50 years time.

Once again many thanks.

Reviewer 2 Report

Dear author, the presented work is very interesting and rich in data. The applied method is sound, and the conclusions are interesting.

As general comment, I have to suggest, unavoidably, to add to your nice work some data from 2020 and 2021 and, even briefly, comment about how the COVID-19 pandemic has changes the EWM in the last two years with respect to the previous years. In particular, it would be interesting to compare it with the Spanish flu.

More in general, adding more details about the applied analysis procedures would help the reader, in particular
for readers that are not experts in th field, since this paper may attract the attention of multidisciplinary researchers.
For instance, when you apply scale factors or other corrections, it would be helpful to provide formulae that define the
applied scaling/corrections, etc.

In some cases, the text and the presentation of data in plots may be improved, as suggested in specific coments below:

  • line 30: please, define more precisely what you intend as "rolling total", better with a formula, in order to avoid possible ambiguous interpretations.
  • lines 73-74: the sentence is not very clear. Please, define more precisely what you have done for those years, and what you intend as "typical summer month totals": "Missing monthly data for the five years 1905 to 1909 in the US time series was reconstructed from annual totals using typical summer month totals either side of these years as a baseline."
  • lines 85-87: please elaborate more about your approach to adjust all contries to the USA, and why is it needed. Also, I would suggest to comment about the magnitude of applied scale factors: "Given that all EWM values in this study have been adjusted to USA-equivalent EWM, individual country values below 5% have been adjusted to 5% while values above 20% have been adjusted to 20%."
  • sec. 2.4: please, elaborate more about the extrapolation of vaccine rates. It is not very clean what procedure has been applied
  • Fig.1: was Denmark scaled w.r.t USA in order to compare the two trends? If so, what was the scale factor applied to Denmark? I think it is worth reporting those information in this part of the text
  • Fig. 2: just for a more clear and immediate presentation, I would suggest to change the vertical axis labels from 1, 2, ..., 6 to Jan, Feb, Mar, May, Jun, Jul. This would also allow you skip the explanation in the caption.
  • Fig. 3: the overlapping curves make the chart rather hard to follow. I would suggest to add another plot which has on the horizonatl axis the year, as in Fig. 3, and on the vertical axis the percentage of deaths in each momth (for this plot the rescaling in order to be adjusted to equal days per month, would not be strictly needed), presented as a stack plot, so that the maximum in each year is always 100%, and each month is represented as a colored band. Equal death for every month would be represented as 100%/12 =8.3% thick band; excess or deficit would result in a larger or smaller values, whose historic trend can be visually followed along the horixontal direction. An example of such chart is showed here:
    https://www.anychart.com/products/anychart/gallery/Area_Charts/100_Percent_Stacked_Area_Chart.php
  • Fig. 4: the chart is very hard to follow. I would suggest to replace the lines by a colored band between two extreme curves. A lighter area for the min-to-max band, a darker area for the lower-to-upper quartile, and a dark line for the median. This is an example:
    https://stackoverflow.com/questions/53071379/using-percentiles-of-a-timeseries-to-set-colour-gradient-in-pythons-matplotlib
  • lines 220-221: please provide more details about the mentioned "adjustment for the effect of cold temperature extremes" that is used in the three provided references.
  • Fig. 5: all three lines show an evident correlation, but have a negative intercept. Does this allow to conclude that there is abaseline EWM around 8% that is present even with no contribution from influenza? I think this would be an interesting conclusion to remark in the text.
  • lines 242-243: "A slight trend to higher EWM was seen, however, there was no statistical difference from the null hypothesis, namely, no change". Can you quantify the statistical significance of the discrepancy of the two hypotheses with a p-value, given one of the test statistics you are using?
  • line 247-248: same as above.
  • line 262: I would avoid the work "postulated". Probably it would be safer to write: "we presume that EWM may be trending upward ..." or "we conclude that there is indication that..."
  • line 266: please, define in the text the acronym "VE", that is not present.
  • Fig. 7: please describe in more details in the text how you include hte "effect of influenza vaccination", which is not very clear.
  • Fig. 7: a 5th order polynomial may overfit the data, given the very sparse distribution of EWM over years. I would add a caveat warning in the text, saying that this cannot be trusted to be a realistic model of the trend.
  • Supplemental material: I think more details about the meaning of the various adjustment factors reported in supplemental material is needed, in order to make them more useful to the reader. This could be done by adding some formulae that relate USA data to other contry's data. It is not clear to me if scale factor are directly applied to EWM or some more elaborate correction is applied.
  • Fig. S1: please, provide in the text more details about the presented data. In particular, what is "Suggested flu deaths" and how should be compared with the "new" and "old" CDC model.
  • Fig. S2: how did you chose 4th or 5th orer polynomial for different countries? Why not using the same choice for all four cuntries?

Author Response

Many thanks for your time and valuable comments.

Please note additional data from Sweden has been included in Figure 1 and data from the US states regarding obesity enabled an estimate of the contribution from obesity into the EWM trends.

COVID-19 section duly added. Were COVID to have occurred back in 1918/19 who can tell what may have happened? Best estimate is that Spanish flu was worse than COVID. This is without going into great detail about the very high level of cytokine storm in Spanish flu which I supect made the big difference.

More details regarding the methods has been added and the scaling factors explained. The scaling factors do change slightly when the median EWM is calculated for different time ranges. This has been made clear. However, still the best available tool to adjust all countries to the US equivalent. Scaling not required in Figure 1 as all 3 countries have a similar median EWM.

Line 30: terminology changed to rolling/moving total with additional description of the calculation.

Lines 73-74: an average was calculated for each month using years either side. The annual total was scaled accordingly with excess deaths added to the winter months.

Lines 85-87: Additional comments added to explain.

sec. 2.4: Additional comments have been added.

Fig 1: No adjustment required. Explained in text with median EWM shown.

Fig 2: Alas cannot get Excel to display the months as text.

Fig 3: There are probably already too many charts in this data rich study. Shall we leave this in its current form. I did try multiple mays of showing the data but came back to this chart in the end. It is there to visually establish the concept.

Fig 4: Alas only have Excel available. 

Figure 5 and Figure S1: Important point. Have added a new section to explain.

Figure S2: Rationale explained.

Supplemental material. Adjustment factor formula added in document

Figure 7: Explanation added to both points

Acronimym for VE, vaccine effectiveness added before VE

Line 262, postulated changed to presumed

Lines 242-243, 247-248. T statistic added.

Lines 220-221: explanation added

Round 2

Reviewer 2 Report

Dear authors,

thanks for taking into account my comments. There are basically two requested points which you didn't consider for addition:

  1. I suggested to extend the study to 2020-2021 data and present the effect of COVID-19 pandemic. There are publicly available data, I can't estimate how much effort it would take to include those additional data. I belive that this addition would substantially increase the interest in this paper.
  2. I suggested a way to visually improve the presentation of some of the plots. It is a pity that technical limitations with excel don't allow a better rendering of your plots. I still believe it is not a big deal to implement the improvement I suggested. Please, find below a possible solution, which I believe will not take too long to implement:
    https://superuser.com/questions/233027/fill-an-area-between-two-lines-in-a-chart-in-excel

Anyway, I leave the editor to judge if the two suggestions are mandatory or the paper can be published as it is.

For the remaining part of the manuscript I consider your replies satisfactory.

Author Response

Suggestion 1 was in fact implemented and the charts included both the first and second winter of COVID-19. An additional paragraph was added to explain.

May we please leave the figure as it currently stands. It is a minor part of the paper and shading between the lines is unlikely to be of vast benefit. However, thank you for suggesting the shading.